# REPOFIXEVAL: A REPOSITORY-LEVEL PROGRAM REPAIR BENCHMARK FROM ISSUE DISCOVERING TO BUG FIXING

## ABSTRACT

Automatic Program Repair (APR) aims to automatically fix software bugs, playing an essential role in software development. While current research demonstrates that Large Language Models (LLMs) excel in file-level program repair, their effectiveness in repository-level program repair remains unexplored. Real-world software projects, which often consist of multiple files, present significant challenges for LLMs in identifying bugs and generating fixes due to the intricate project structures. To bridge this gap, we introduce REPOFIXEVAL, a repository-level APR benchmark consisting of 160 real-world bug fixing suites from popular Python projects. REPOFIXEVAL provides the original buggy programs, associated issue reports, corresponding fixes, and unit tests to verify the correctness of each fix. Based on the benchmark, we further propose a three-step evaluation framework for LLM-based APR tools, encompassing (1) discovering issues from execution failures, (2) localizing buggy code segments, and (3) generating code fixes. Experimental results highlight that LLMs struggle with organizing error messages during the issue discovery phase. We find that longer contexts positively affect performance, but only a few LLMs can effectively utilize extended context information at the 128K level. Some open-source LLMs demonstrate competitiveness with closed-source counterparts, yet even the best-performing GPT4-o only resolves 12.3% of bugs. Our study reveals the capabilities and limitations of 16 LLMs in handling repository-level bugs, providing valuable insights for future research in this field.

## 1 INTRODUCTION

Automated Program Repair (APR) aims to automatically detect and generate code fixes (i.e., patches) for a given piece of buggy code. APR is crucial in enhancing the overall reliability of software systems by accelerating the debugging process and reducing the chances of human error. Over the years, APR tools have evolved from conventional rule-based methods (Huang et al., 2024) to contemporary data-driven, intelligent approaches. Recent advancements in large language models (LLMs), such as GPT-4 (OpenAI, 2023) and Llama3.1 (Dubey et al., 2024) offer alternative solutions for more complicated program bugs without relying on historical code patches.

Despite the numerous efforts in developing APR approaches, their evaluation is limited in an over-simplified scenario of **function-level or file-level** bugs (e.g., DebugBench Tian et al. (2024)). Consequently, there remains a gap between these benchmarks and real-world complicated development environments, where developers often deal with multi-file projects. Resolving project-level program bugs presents substantial challenges in comprehending the relationships between functions across multiple files and localizing faults within extensive codebases.

To fill this gap, in this paper, we propose a **practical repository-level** program repair benchmark, named REPOFIXEVAL, consisting of 160 repository-level program bugs from 16 GitHub projects. These bugs are selected from widely-used repositories with executable unit tests in high coverage, ensuring their quality and reproducibility. Five experienced programmers further manually identify crucial bug fixes from commit histories. As a result, REPOFIXEVAL collects buggy programs,

| Method | Repo Level | Test Cases | Fault Localization | Issue Raised | Issue Evaluation |
|---|---|---|---|---|---|
| DebugBench (Tian et al., 2024) | ✗ | ✓ | ✗ | ✗ | ✗ |
| EvalGPTFix (Zhang et al., 2023) | ✗ | ✓ | ✗ | ✗ | ✗ |
| FixEval (Haque et al., 2023) | ✗ | ✓ | ✗ | ✗ | ✗ |
| HumanEval-Java (Jiang et al., 2023) | ✗ | ✓ | ✗ | ✗ | ✗ |
| Review4Repair (Huq et al., 2022) | ✗ | ✗ | ✓ | ✗ | ✗ |
| QuixBugs (Lin et al., 2017b) | ✗ | ✓ | ✗ | ✗ | ✗ |
| ∞Bench (Zhang et al., 2024) | ✓ | ✗ | ✓ | ✗ | ✗ |
| SWE-bench (Jimenez et al., 2023) | ✓ | ✓ | ✗ | ✗ | ✗ |
| RepoBugs (Chen et al., 2024) | ✓ | ✗ | ✗ | ✗ | ✗ |
| REPOFIXEVAL | ✓ | ✓ | ✓ | ✓ | ✓ |

Table 1: A comparison of our REPOFIXEVAL with some notable datasets.

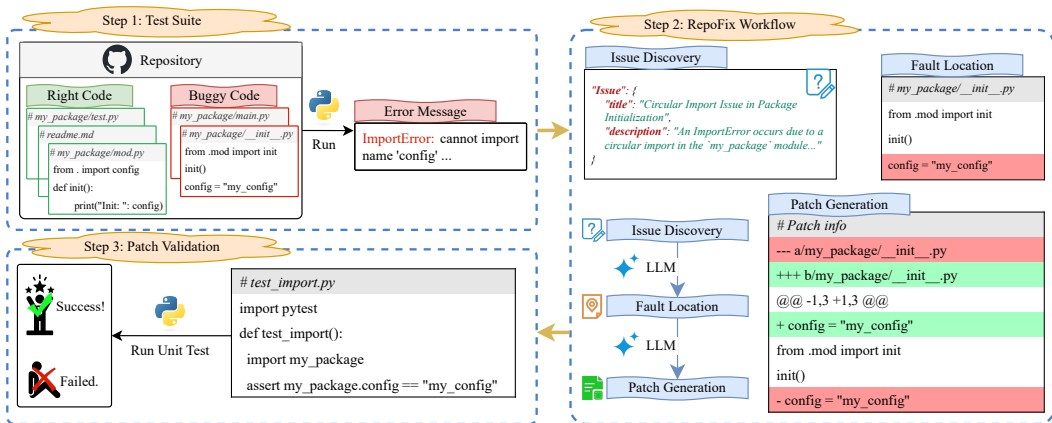

Figure 1: The overview of REPOFIXEVAL. Given a repository with buggy code, possibly with error messages, the repository and error messages are input into LLMs. The Repofix workflow then generates a patch. A fix is considered successful only if the patch passes all unit tests.

related issue reports, unit tests, and corresponding patches (i.e., fixes) for each bug. The differences between REPOFIXEVAL and the previous benchmarks are shown in Table 1.

REPOFIXEVAL introduces a novel **issue-aware** scenario for program repair, inspired by bug-reporting and bug-resolving processes in real-world software development. Rather than fixing "existing well-described error reports" from clients, as in SWE-Bench (Jimenez et al., 2023), this setting requires LLMs to autonomously identify and summarize the bug symptoms based on current information. As illustrated in Fig. 1 (Step 2), given a buggy repository and associated runtime error messages, the program repair workflow involves three steps: (1) discovering issues from execution results, (2) localizing faults within the codebase, and (3) producing patches to fix the bug.

To systematically understand how existing LLMs can resolve repository-level program repair problems, we assess their performance on each of the aforementioned steps. First, for issue generation, we use the "LLM-as-a-judge" framework (Zheng et al., 2023a) to assess the reproducibility, relevance, explanation, and overall of the produced issues. Second, we measure the accuracy of the identified faulty lines against the actual ones for fault localization. Third, we evaluate the bug patches by verifying whether the patched code passes all unit tests.

We evaluate 16 LLMs, systematically examining the impact of context lengths ranging from 2,000 to 128,000 tokens on performance, and set 14 different input settings for each length. When provided with error messages, GPT-4o achieved the highest patch pass rate of 11.25%. Notably, we observed that most LLMs demonstrated performance improvements within a context length of 16K tokens. However, only selected LLMs maintained their performance capabilities at extended context lengths of up to 128K tokens. Through rigorous quantitative analysis, we established significant correlations among the three key tasks: issue discovery, fault localization, and patch generation.

The contributions are summarized as follows:

- **A Repository-Level APR Evaluation Dataset:** We propose a novel repository-level APR benchmark, named REPOFIXEVAL. We imitates real-world project-based software practices, bridging the gap between existing single-function bug datasets and complicated development needs.
- **Comprehensive Evaluation Framework:** We design a thorough evaluation framework encompassing issue discovery, fault localization, and patch generation. To the best of our knowledge, this is the first framework to focus on summarizing issues from execution results, presenting a more challenging yet practical scenario for program repair.
- **Comprehensive LLM Evaluation and Analysis:** Empirical evidence on the performance of 16 state-of-the-art LLMs across issue discovery, fault localization, and patch generation tasks, revealing their strengths and limitations. We further conducted an analysis of the impact of context length and input setting on task performance, as well as the correlations between tasks.

## 2 RELATED WORK

**LLMs for Automated Program Repair.** Automated Program Repair (APR) seeks to assist developers by automatically generating patches for software bugs (Gazzola et al., 2018). APR tools produce patched code based on the original code after identifying buggy locations (Fan et al., 2023; Wei et al., 2023; Ye et al., 2022a). There are three traditional APR tools types: template-based (Koyuncu et al., 2020), learning-based (Ye et al., 2022b) and LLM-based methods. While template-based tools require human-intensive labour to define fixing rules, learning-based tools can generate more versatile and expressive patches after training on historical bug-fixing data. More recently, LLMs such as Starcoder (Li et al., 2023), WizardCoder (Luo et al., 2023), mCoder (Chai et al., 2024) and UniCoder (Sun et al., 2024) have shown potential for APR due to their exceptional performance and remarkable generalization capabilities across a diverse range of code-related tasks (Prenner et al., 2022).

**Benchmarks for Automated Program Repair** Popular APR benchmarks like Defects4j (Just et al., 2014), QuixBugs (Lin et al., 2017a), DebugBench (Tian et al., 2024) and EvalGPTFix (Zhang et al., 2023) enable the evaluation of code LLMs' auto repair ability but mainly focus on function-level or file-level tasks. CrossCodeEval (Ding et al., 2023) and RepoBench (Liu et al., 2023) have introduced cross-file tasks to address this limitation, but they only focus on code completion tasks.In the realm of repository-level code repair, SWE-Bench (Jimenez et al., 2023) stands out by focusing on resolving issues present in existing GitHub repositories. Nevertheless, its scope is confined to human-reported issues, lacking an evaluation of large language models' ability to discover issues autonomously and fault localization. ReposVul (Wang et al., 2024) has only introduced a repository-level vulnerability dataset. Similarly, while RepoBugs (Chen et al., 2024) has delved into repository-level code fix tasks, its evaluation methodology primarily emphasizes the final outcome rather than the entire issue identification and code localization process. Moreover, RepoBugs relies on human experts to judge the correctness of code modifications, lacking an automated evaluation mechanism.

## 3 REPOFIXEVAL: A BENCHMARK FOR REPOSITORY-LEVEL CODE FIX TASK

REPOFIXEVAL is a benchmark containing repository-level program bugs. The task is to discover issues, localize faults, and ultimately generate a patch that resolves the bug to pass the relevant tests.

### 3.1 BENCHMARK CONSTRUCTION

We construct REPOFIXEVAL based on the following principles to ensure its representativeness of bug-fixing practices: (1) Code should originate from real-world and active projects; (2) The project should contain mature test suites for easier maintenance; and (3) Bug cases should be well-annotated and of high quality.

**Stage I: Real-World Code Repository Collection.** Similar to (Li et al., 2024) and (Jimenez et al., 2023), we searched GitHub for high-quality open-source repositories, focusing on those with

well-known organizations and multiple source files to represent mainstream development environment. We chose Python repositories due to its popularity and robust testing frameworks.

**Stage II: Execution-based Filtering.**   We built a runtime environment for each repository to execute unit tests, ensuring the functionality of the code. Repositories without automated test suites (e.g., `pytest`, `unittest`) or with failed tests were excluded to guard the reproducibility of unit tests in our framework.

**Stage III: Annotation & Quality Control.**   Five experienced software developers annotated issues using two methods: (1) identifying fix-related commits from git history, and (2) manually injecting bugs to simulate real-world scenarios. Developers proposed issues and solutions for each bug. Solutions that pass all unit tests are considered correct. An expert reviewed each case for quality control. The final test case included a buggy codebase, issues, solutions, bug localization, unit test code, and commands. For each case, issues were proposed from the user's perspective, and solutions were provided from the developer's perspective, serving as reference answers.

In summary, REPOFIXEVAL contains 160 bug cases, covering projects from natural language processing, machine learning, database applications, algorithm implementation, and general tools. Table 2 shows statistics of the benchmark.

| **Repo Statistics** | Star | Fork | # File | File_size (KB) |
|---|---|---|---|---|
| | 351.08 | 91.56 | 27.75 | 514.15 |
| **Dataset Statistics** | # File | # Modify_line | # Patch_token | # Issue_token |
| | 1.91 | 10.54 | 256.26 | 210.34 |

Table 2: Benchmark Statistics. The Statistics represent the mean results averaged over each repository and individual bug case.

## 3.2 Evaluation Tasks and Metrics

### 3.2.1 Task 1: Issue Discovery

An issue refers to a user-created report of a problem or bug, including a title and a detailed description of the problem and steps to reproduce the bug. When users encounter bugs or identify potential improvements, they can create such issues in an issue tracking system (e.g., GitHub, Jira) to notify project maintainers. In this task, the model is asked to simulate the issue creation process based on bug symptoms.

**Model Input & Output**   Given code repository $\mathcal{C}$ and the error message from unit tests $\mathcal{E}$, LLMs $\mathcal{M}$ are required to generate the corresponding issue $i = \mathcal{M}_{issue}(\mathcal{C}, \mathcal{E})$. The output issue $i$ is structured in JSON which has `title`, `description` and `explanation`, ensuring a standardized representation that can be easily disseminated among development tools and teams.

**Evaluation Metrics.**   Given the open-ended nature of proposing and evaluating issues, we employ the LLM-as-a-judge (Zheng et al., 2023a) method as the standard for evaluation. For the given issues, we primarily assess them from four perspectives: reproducibility $I_r$, relevance $I_v$, explanation $I_e$ and overall $I_o$. Reproducibility ($I_r$) assesses the presence and quality of step-by-step instructions, scripts, or code references for reproducing the issue. Relevance ($I_v$) evaluates the accuracy in identifying the bug, linking the problem to error messages, and providing an effective solution. Solution explanation ($I_e$) examines the clarity and completeness of the proposed solution, including its rationale and validation. Overall quality ($I_o$) provides a holistic assessment of the issue description's comprehensiveness and clarity.

Formally, let $i, \mathcal{E}$ represent model-generated issues and error messages respectively, and let $\mathcal{P}$ denote the patch that can fix the bugs. We set the reference score for an answer as 8 points. $I_x$ (where $x \in \{r, v, e, o\}$) is evaluated on a scale from 0 to 10: $I_x = \mathcal{M}_{eval}(i, \mathcal{P}, \mathcal{E})$ The overall evaluation score $I_{Avg}$ can be defined as $I_{Avg} = \frac{1}{4}(I_r + I_v + I_e + I_o)$. To align with the subsequent evaluation metrics, we multiplied the value of $I$ by 10, resulting in a score range of 0 to 100.

### 3.2.2 Task II: Fault Localization

The subsequent step, fault localization, aims to identify the exact code lines for a particular defect. Pinpointing the precise segment of code that leads to a specific issue requires not only a deep understanding of the codebase but also an ability to effectively navigate the directory structure.

**Model Input & Output** For the current code repository $\mathcal{C}$, corresponding issue $i$ and error message $\mathcal{E}$, we prompt LLMs to pinpoint the code segments $l$ requiring modification: $l = \mathcal{M}_{loc}(i, \mathcal{C}, \mathcal{E})$ In fault localization, we define two location granularities: (1) **Filename**: Identifies the specific file containing the error, serving as the initial debugging step. (2) **Specific Code**: Determining the exact line or code snippet needing change ensures targeted modifications, formatted as *'line number: content of the current line'*.

**Evaluation Metrics** To evaluate the performance of LLMs in the fault localization task, we employ a set of comprehensive and multidimensional metrics. These metrics are designed to assess different aspects of the generated outputs, including the accuracy of the JSON schema, filenames and specific code content.

- If the model's output is in the correct JSON format, we will use F1, precision, and recall metrics for evaluation. We integrate the evaluation content into the form $\{\texttt{ID}\}\char`^\{\texttt{filename}\}\char`^\{\texttt{line number}\}$, where ID represents the task identifier, filename is a string, $\char`^$ is the delimiter, and the line number indicates the specific lines of code to be modified. Each line requiring modification constitutes a separate evaluation case, collectively forming a set. The ground truth set is denoted as $S_{gt}$, and the model output set is denoted as $S_{pd}$. We use the proportion of correctly predicted variables (precision $= \frac{|S_{pd} \cap S_{gt}|}{|S_{pd}|}$), the proportion of actual variables predicted by the model (recall $= \frac{|S_{pd} \cap S_{gt}|}{|S_{gt}|}$), and their harmonic mean (F1 $= 2 \cdot \frac{\text{precision} \cdot \text{recall}}{\text{precision} + \text{recall}}$).

- Additionally, we evaluate whether the model's output can be parsed into JSON, termed $\texttt{Loads@k}$. This means that the model generates K samples, and if any sample can be successfully parsed, it is considered a successful parse. Finally, the proportion of successful parses is calculated.

### 3.2.3 Task III: Bug Fixing

Unit testing is crucial for detecting errors in pre-deployment period, significantly reducing the effort required to address bugs in later stages. REPOFIXEVAL amis to generate a code patch that can fix the identified bug while ensuring all unit tests pass and original functionality is maintained.

**Model Input & Output** After issue discovery and fault localization, we focus on effective code repair. Inspired by SWE-Bench, we use LLMs to generate patch files instead of complete code. Given the code repository $\mathcal{C}$, issue $i$, fault localization $l$, and error message $\mathcal{E}$, we prompt LLMs to produce patch $p$ to resolve the bug: $p = \mathcal{M}_{patch}(i, l, \mathcal{C}, \mathcal{E})$

**Evaluation Metrics** We employ a comprehensive evaluation process to assess patch effectiveness. Initially, we verify the patch's correctness by integrating it into the codebase without conflicts using Git. Next, we run unit tests to confirm that the patch successfully resolves the bug.

To prevent potential exploitation where LLMs might modify test code to pass evaluations, we implement an additional two-step verification process: *(1) Code Segregation*: We separate the codebase into functional code and test code. *(2) Dual Testing*: After applying the patch to the functional code, we conduct tests using both original original (pre-patch) test code and potentially modified (post-patch) test code. A patch is deemed valid only if it passes all tests in both scenarios, ensuring that the fix is genuine and not a result of compromised test code. This rigorous approach guarantees that patches not only resolve the initial issue but also maintain overall system integrity.

# 4 EXPERIMENTS

## 4.1 EXPERIMENT SETUP

**Code LLMs.** We evaluate 16 instruct models with sizes ranging from 1.5B to 405B parameters, including open/closed-source LLMs. For general models, we evaluate GPT-4o-2024-05-13, GPT-4o-mini-2024-07-18 (OpenAI, 2023), Llama3.1 (Dubey et al., 2024), Qwen2 (Yang et al., 2024) and GLM-4 (GLM et al., 2024). For code models, we evaluate Qwen2.5-Coder (Hui et al., 2024), DeepSeek-V2.5 (Zhu et al., 2024), Codestral (mistralai, 2024) and CodeGeeX4 (Zheng et al., 2023b).

**Code Retrieval.** Given that the content of most code repositories significantly exceeds the context length limit supported by the models, it is infeasible for the models to read the entire repository content at once. Therefore, we employ the BM25 (Robertson et al., 2009) retrieval algorithm to retrieve relevant files. To include potential buggy code in the input for LLMs, we extracted relevant code segments according to the error messages for each case and referenced issues.

**Implementation Details.** Our experimental setup utilizes 8 NVIDIA A100 (80G), leveraging the vLLM (Kwon et al., 2023) as the inference backend. For the DeepSeek and GPT-4 models, we employed the official APIs. To ensure the diversity of issue generation, we applied a temperature setting of 0.95 and a top-k sampling of 0.2. To validate the ability of LLMs to correctly generate JSON format, we adopted `Loads@3`. This process verifies the model's output up to three times to ensure conformity with JSON format. If the output fails to be parsed by `json.loads` in Python after the third attempt, we retain the model's original output.

**Input Settings for Evaluation** To explore LLM's ability on code analysis, we input the codebase with specific prompts to discover potential bugs. Additionally, we consider errors generated during the execution of unit tests. By inputting the codebase and associated error messages with comprehensive prompts, we guide LLMs to identify potential bugs. We also require LLMs to output explanations and solutions from a developer's perspective for each issue. After discovering issues, we prompt LLMs to perform fault localization, analyzing each issue to determine its specific fault location within the codebase. To further verify localization capabilities, we input reference issues and their solutions. Given a codebase and its issues, we prompt LLMs to generate patches to resolve identified bugs. To evaluate the impact of localization information, we use two methodologies: one with only the issues and another with issues and their localization information. Additionally, we include reference issues and localization information for comprehensive evaluation. As shown in Table 3, we use various input settings to evaluate LLMs.

| Task Type | Symbol of Output | Input Setting |
|---|---|---|
| Issue Discovering | $i_{origin}$ | Pure Repository |
| | $i_{message}$ | Pure Repository and Error Message |
| | $i_{oracle}$ | Reference issues, only generate explanation |
| Fault Localization | $l_{origin}$ | From $i_{origin}$ |
| | $l_{message}$ | From $i_{message}$ |
| | $l_{oracle}$ | From $i_{oracle}$ |
| | $l_{oracle\_exp}$ | Reference issues and explanations |
| Patch Generation | $p_{origin}$ | From $i_{origin}$ |
| | $p_{message}$ | From $i_{message}$ |
| | $p_{origin\_loc}$ | From $i_{origin}$ and $l_{origin}$ |
| | $p_{message\_loc}$ | From $i_{message}$ and $l_{message}$ |
| | $p_{oracle}$ | From $i_{oracle}$ |
| | $p_{oracle\_location}$ | From $i_{oracle}$ and $l_{oracle}$ |
| | $p_{oracle\_exp}$ | Reference issues and explanations |

Table 3: Input Settings for Issues, Localization, and Patches.

| Model | Size | Issue Task | | Location Task | | Patch Task | |
|-------|------|------|------|------|------|------|------|
| | | Rep | Avg | Loads | F1 | Apply | Pass |
| GPT-4o | 🔒 | 56.13 | **67.32** | 94.38 | **27.02** | 18.75 | **11.25** |
| DeepSeekV2.5 | 236B | **58.13** | 67.15 | 91.88 | 10.87 | **19.38** | 9.38 |
| GPT-4o-mini | 🔒 | 51.53 | 63.64 | **100.00** | 19.28 | 5.00 | 1.88 |
| Meta-Llama-3.1-405B | 405B | 47.56 | 59.63 | **100.00** | 19.64 | 6.25 | 1.88 |
| Codestral-22B-v0.1 | 22B | 28.50 | 34.36 | 36.25 | 7.17 | 3.13 | 1.25 |
| Qwen2.5-Coder-7B | 7B | 28.13 | 35.01 | 50.63 | 6.37 | 5.00 | 1.25 |
| CodeGeeX4-All-9B | 9B | 44.94 | 54.68 | 86.88 | 10.48 | 1.88 | 0.63 |
| GLM-4-9B | 9B | 47.06 | 56.71 | 93.13 | 8.41 | 1.88 | 0.63 |
| Meta-Llama-3.1-70B | 70B | 39.09 | 54.31 | 98.75 | 13.73 | 3.13 | 0.63 |
| Meta-Llama-3.1-8B | 8B | 43.56 | 55.41 | 85.63 | 7.22 | 2.50 | 0.63 |
| GLM-4-9B-1M | 9B | 43.50 | 54.62 | 84.38 | 5.52 | 13.75 | 0.00 |
| Qwen2-72B | 72B | 29.94 | 35.79 | 52.50 | 11.07 | 0.63 | 0.00 |
| Qwen2-7B | 7B | 28.94 | 35.60 | 45.63 | 8.94 | 0.00 | 0.00 |
| Qwen2.5-Coder-1.5B | 1.5B | 19.94 | 29.70 | 42.50 | 6.28 | 1.25 | 0.00 |
| Qwen2-57B-A14B | 57B | 28.63 | 36.50 | 41.25 | 7.10 | 0.63 | 0.00 |
| Qwen2-1.5B-Instruct | 1.5B | 18.38 | 26.80 | 0.63 | 0.00 | 0.00 | 0.00 |

Table 4: Main results of REPOFIXEVAL, where *Rep* refers to reproducibility in issue discovery task. The input context lengths for Qwen2-57B-A14B and Qwen2-1.5B-Instruct are 64K and 32K tokens, respectively. The results are sorted in descending order by the `Pass` metric in the Patch Task.

## 4.2 MAIN RESULTS

Our comprehensive evaluation of LLMs on the REPOFIXEVAL repository-level code fix task reveals several key insights into their performance across issue discovery, fault localization, and bug fixing. As depicted in Table 4, we verify the performance of the model at the 128K token level based on the provided message information($i_{message}, l_{message}, p_{message\_loc}$).

Figure 2 provides a holistic view of model performance across representative metrics in issue discovery (e.g., `Avg`, `Rep`), fault localization (e.g., `Loads`, `F1`), and patch generation (e.g., `Apply`, `Pass`). The radar chart illustrates that while some models excel in certain steps in program repair, no single model dominates the complete process. Given that the score is on a 100 grading scale, it is evident that current models still perform unsatisfactory on both localization and patch generation tasks. The performance discrepancy between different tasks reveals the complexity of the repository-level debugging problem and suggests the need for multi-task evaluation.

In issue discovery, GPT-4o and DeepSeek-V2.5 demonstrate superior performance with `Avgs` scores of 67.32 and 67.15, respectively. These high scores indicate their proficiency in generating reproducible and well-structured issues. High reproducibility in issue discovery is crucial as it enables developers to consistently recreate the reported problems, facilitating more efficient debugging processes and minimizing false positives. Moreover, GPT-4o leads with a `F1` score of 27.02 in fault localization, reflecting its robust understanding of directory architecture.

In patch generation, the performance of LLMs shows a significant decline compared to their effectiveness in issue discovery and fault localization, highlighting the challenges of resolving the issue without introducing new bugs. GPT-4o also excels with the highest "`Pass`" score of 11.25. DeepSeek-V2.5 follows closely, demonstrating significantly outperforming GPT-4o-mini, demonstrating that open-source models have the potential to surpass closed-source models. Notably, some models, such as GLM-4-9B-Chat-1M, exhibit a significant discrepancy between the percentage of applicable patches (13.75%) and the percentage of correct patches after passing all tests (0%).

## 5 FURTHER ANALYSIS

### 5.1 CORRELATION ANALYSIS

We further explore the effects of upstream tasks on the subsequent tasks in program repair in Fig. 3. Specifically, we concentrate on whether a LLM excelling in one upstream task (e.g., issue discovery)

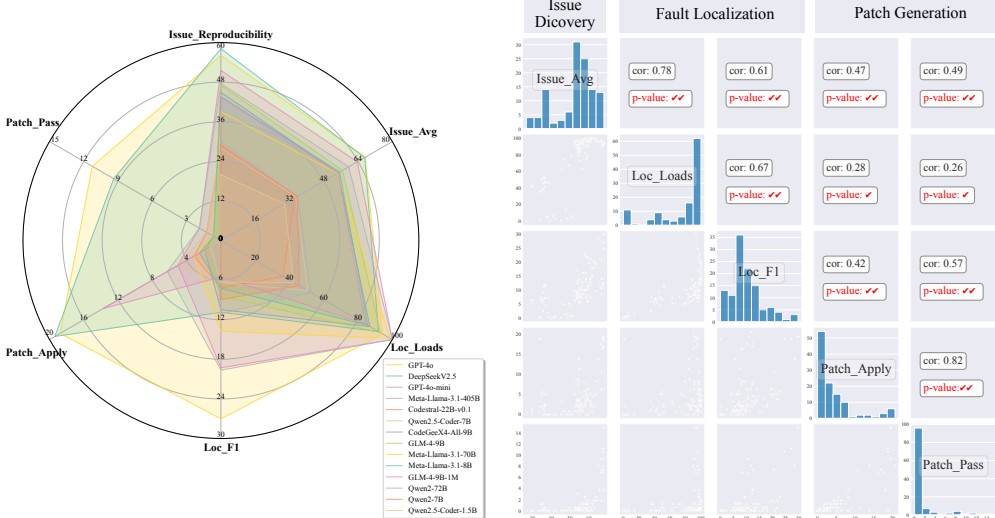

Figure 2: Radar chart depicting the performance on issue discovery, fault localization, and patch generation.

Figure 3: Pair Plots of five key performance indicators across three tasks.

will result in better results in the downstream one (e.g., patch generation). Experiment results in Fig. 3 include all data from the $(i_{message}, l_{message}, p_{message_{loc}})$ task, with token counts ranging from 2K to 128K. As shown in Fig. 3, the histograms on the main diagonal display the distribution of each evaluation metrics. The lower triangular portion presents scatter plots between different evaluation metrics, while the upper triangular portion indicates the Pearson correlation coefficient (cor) and its p-value for each pair. A single checkmark (✓) denotes significant correlations where the p-value is less than 0.01, while two checkmarks (✓✓) indicate highly significant correlations with p-values less than 0.001.

We observed that all metrics demonstrate statistical significance. Notably, the quality of issues (as measured by Issue_Avg) exhibits strong correlations with all other task metrics. In particular, there is a pronounced relationship between issue quality and Loc_Loads. This loading rate, in turn, influences the F1 score Loc_F1. Furthermore, the application rate Patch_Apply has a substantial impact on the final pass rate Patch_Pass in the patch generation task.

## 5.2 CONTEXT LENGTH

To investigate the impact of context length on model performance, we selected 11 LLMs that are capable of supporting up to a 128K context length. In specific, we extracted text segments using code retrieval (Section 4.1) with lengths of 1,500, 3,000, 6,500, 13,000, 30,000, 61,000, and 124,000 tokens, corresponding to the context limits of LLMs with 2K, 4K, 8K, 16K, 32K, 64K, and 128K token, respectively. The experimental results for various context lengths are shown in Fig. 4.

**Observation in Issue Discovering.** First, the issue discovering performance of most LLMs slightly improves or remains relatively stable as context length increases, particularly a noticeable improvement within shorter context lengths up to 16K. However, after the 16K context, Qwen2.5 and Codestral-22B-v0.1 show a significant performance decline, which may be caused by the distribution of training data. This decline is likely due to insufficient long-text training data during the training stage. Besides, some open-source models like DeepSeek-V2.5 perform competitively with commercial models, even surpassing GPT-4o at certain context lengths.

**Observation in Fault Localization.** As the context length increases, most LLMs initially show performance improvement followed by a decline. However, only a few LLMs like Meta-Llama-3.1 maintain performance beyond 16K tokens, with GPT-4o showing a more noticeable upward trend. This indicates that while richer contextual information can enhance performance, only select LLMs

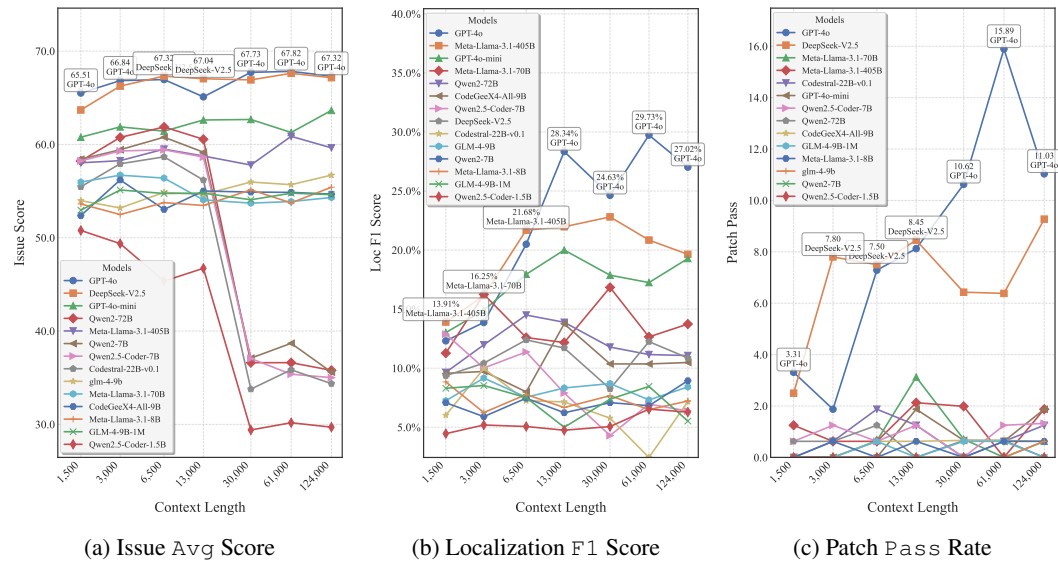

(a) Issue `Avg` Score      (b) Localization `F1` Score      (c) Patch `Pass` Rate

Figure 4: Comparison of Issue `Avg` Score, Localization `F1` Score, and Patch `Pass` Rate vs. Context Length.

effectively utilize extremely long contexts. And the open-source model Meta-Llama-3.1-405B outperforms the closed-source GPT-4o in scenarios with context lengths below 8K, and consistently surpasses GPT-4o-mini across all context lengths. This indicates that open-source LLMs can, to some extent, serve as alternatives to advanced closed-source LLMs.

**Observation in Patch Generation.** Apart from GPT-4o and DeepSeek-V2.5, performance further improved as the context length increased, demonstrating excellent capability in handling long contexts. In contrast, other models performed poorly across all token lengths, with pass rates consistently below 4. This stark difference highlights the inherent difficulty of the patch task and suggests that most LLMs lack the necessary capabilities to effectively utilize contextual information.

### 5.3 Task Input Settings

To investigate the impact of input information on debugging performance, we conducted a total of 14 different task input setting experiments in Table 3 and the result is shown as Fig. 5.

**Feeding error messages does not necessarily lead to the improvement in discovering issues.** The inclusion of error messages appears to improve the reproducibility of model outputs, suggesting that these messages provide crucial context for generating more consistent and verifiable results. However, apart from a few top-tier models like GPT-4o, most models did not show significant overall performance improvements with the help of error messages. In some cases, performance even decreased. This unexpected outcome may be attributed to: *(a) Limitations in Information Organization:* Most models may over-rely on error messages while neglecting the broader context of the issue, resulting in problem descriptions that are more technically accurate (as reflected in higher Reproducibility scores) but less clear in overall expression and structure compared to descriptions without error messages. *(b) Multidimensionality of Evaluation Criteria:* The average score of the issue considers multiple dimensions (Relevance, Explanation and Overall) beyond just Reproducibility. In processing error messages, models may focus excessively on technical details, potentially compromising performance in other aspects.

**Input information quality significantly impacts performance.** There is a clear positive correlation between the quality of input setting and model performance. In fault localization, the condition with reference issues and solutions ($l_{oracal\_exp}$) typically achieves the highest F1 scores, while using error message inputs ($l_{message}$) shows significant improvement over inputs consisting solely of

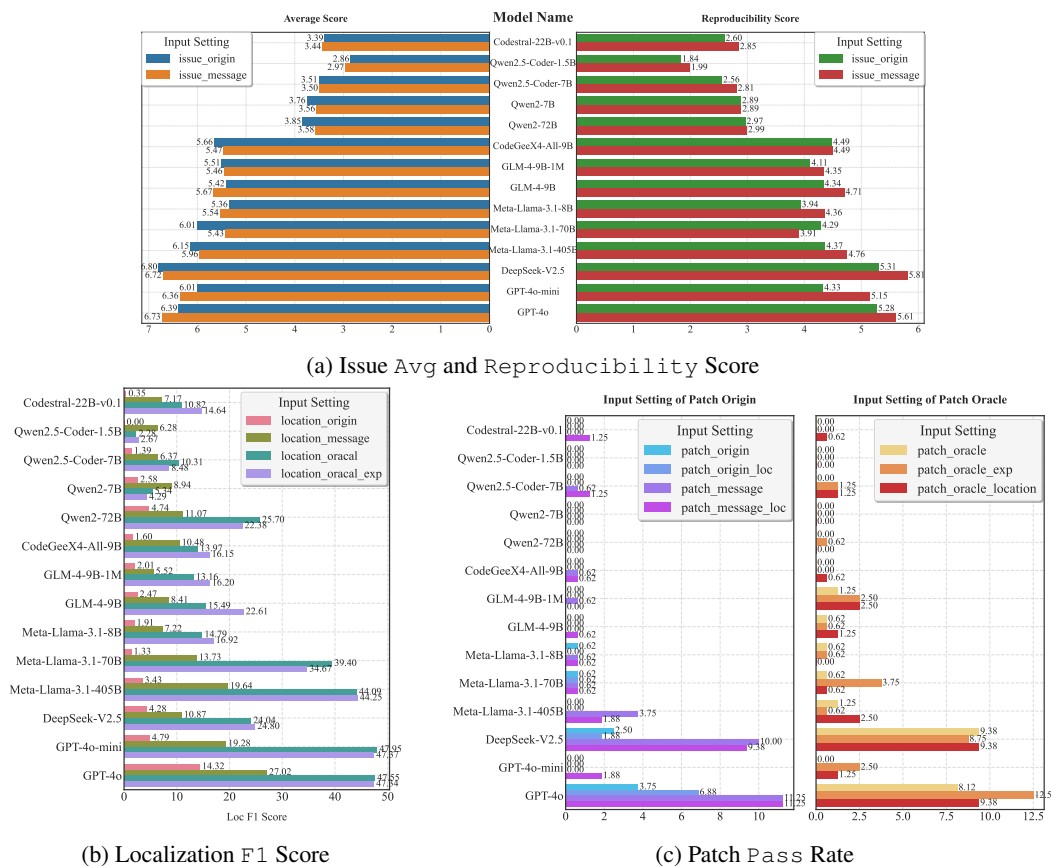

(a) Issue `Avg` and `Reproducibility` Score

(b) Localization `F1` Score

(c) Patch `Pass` Rate

Figure 5: Comparison of Issue `Avg`, `Reproducibility` Score, Localization `F1` Score and Patch `Pass` Rate vs. Input Setting.

the codebase ($l_{origin}$). This highlights the crucial role of error messages in enhancing the fault localization capabilities of LLMs. Similarly, in patch generation, inputs that integrate detailed error messages and precise fault localization demonstrate better results than those relying on the basic codebase. The $p_{message\_loc}$ condition consistently yields the highest pass rate for most models, indicating that the combination of all available contextual information leads to the most effective patch generation. Advanced models like DeepSeek-V2.5 and GPT-4o demonstrate superior performance across all patch types, suggesting a higher capacity to utilize diverse input setting. The enhanced performance of $p_{oracal\_exp}$ and $p_{oracal\_location}$ compared to $p_{oracal}$ alone further underscores the importance of explanations and precise fault localization. Additionally, the performance gap between models with and without oracle information indicates potential areas for improvement in unsupervised patch generation scenarios.

## 6 CONCLUSION

In this work, we propose a novel repository-level APR evaluation dataset, REPOFIXEVAL, created through an annotation and verification process conducted by professional developers. This dataset contains 160 bug suites, including buggy code, comprehensive unit tests, related issues, and bug-fixing patches. We are the first to introduce a full-process evaluation mimicking human bug resolution, where models initially propose relevant issues and subsequently generate patches for solutions. Systematic evaluations of existing LLMs on REPOFIXEVAL reveal performance disparities between open-source and closed-source models. Additionally, our findings highlight current LLM deficiencies in discovering issues, localizing faults, and generating patches, suggesting directions for improvement. This marks a significant advancement for developers using AI techniques to understand and debug effectively in real-world software development environments.

## CODE OF ETHICS AND ETHICS STATEMENT

REPOFIXEVAL is developed using data from public code repositories, ensuring all contributions comply with the respective license requirements. All repositories used in this study comply with the usage requirements of their respective licenses: MIT license, Apache-2.0 license, and BSD-3-Clause license.We do not collect personal information of repository users, and REPOFIXEVAL does not use any data beyond the GitHub public API and website. The selection of repositories for REPOFIXEVAL is based on objective popularity metrics, without involving any discriminatory or biased criteria.

To ensure transparency, we plan to release the REPOFIXEVAL dataset, its collection and evaluation infrastructure and experimental results. Following established best practices, we will provide detailed documentation for all components and establish open communication channels to gather feedback for improving REPOFIXEVAL.

## REPRODUCIBILITY

In our submission, we have uploaded a complete source code archive that has been appropriately anonymized. The source code contains inline documentation detailing the purpose and usage of various parts of the codebase. Additionally, we provide the complete set of REPOFIXEVAL task instances discussed in the paper.

We plan to formally release REPOFIXEVAL as an open-source codebase to the public. This release will include exhaustive details on benchmarking, code structure, and usage instructions. One of the core components of REPOFIXEVAL, the data collection framework, will also be part of the open-source release. Due to its easily maintainable design, as described in the main paper, we believe REPOFIXEVAL will have high reproducibility.

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

## A APPENDIX

### A.1 REPOFIXEVAL'S FEATURE

In contrast to the previous APR benchmarks, which suffer from limitations such as limited scope, lack of end-to-end evaluation, and inadequate performance metrics, our REPOFIXEVAL offers several distinctive features that address these issues comprehensively in Tab.1.

Unlike the above APR datasets, REPOFIXEVAL introduces several distinctive features:

- **Repository-Level Program Repair**: REPOFIXEVAL expands the scope of traditional APR benchmarks by encompassing multi-file projects from real-world repositories. This requires models to identify and repair bugs across various modules and components within a repository, simulating a more interconnected and interdependent software system environment. Unlike conventional datasets that focus on single-file or single-function tasks, this approach provides a more realistic and challenging evaluation framework, highlighting the necessity for models to understand and analyze code spanning multiple files.

- **Offer Unit Test-Based Evaluation**: REPOFIXEVAL utilize unit tests as a primary metric for evaluation. This methodology ensures that the generated patches not only syntactically integrate with the existing code but also maintain functional correctness. By using unit tests, REPOFIXEVAL closely mimics real-world software development practices.

- **Adapt Issue-aware Debugging Pipeline**: REPOFIXEVAL places significant emphasis on the discovery and assessment of issues in the repositories. Just as the issue reports bridge the customer usage failures and the software development team's repair efforts, we have similarly adapted such an issue-aware pipeline for debugging. This feature ensures that the models are capable of identifying and resolving issues, which can demonstrate very well to humans.

These unique characteristics of REPOFIXEVAL aim to bridge the gap between theoretical APR research and practical application, extending the capabilities of APR systems within real-world software development settings. By encouraging models to handle complex, cross-file projects, REPOFIXEVAL contributes to the advancement of more robust and effective APR methodologies.

## A.2 THE DIFFERENCE OF BUGS FROM MULTIPLE STRUCTURAL LEVELS.

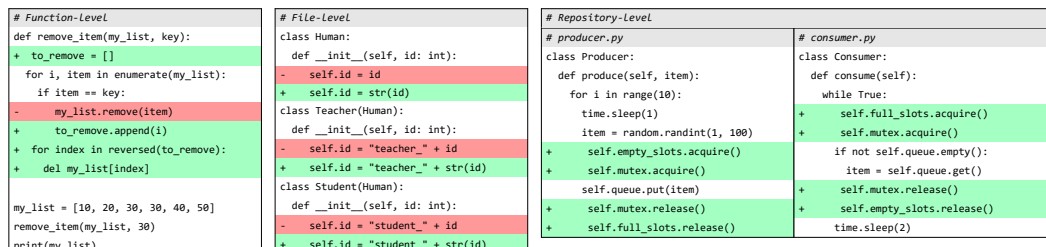

Figure 6: Examples of code bugs across different structural levels.

For instance in Fig. 6, debuggers must comprehend the interrelationships between functions across multiple files and localize faults within extensive codebases.

## A.3 MORE INFORMATION OF THE DATASET

The main types of bugs include logic errors, import errors, reference errors, assignment errors, etc. The average token of error message is 797.28.

The details of repository is in Tab. 5.

## A.4 AN EXAMPLE OF REPOFIXEVAL

An example of REPOFIXEVAL is shown in 1.

| Repository Name | Repository URL |
|---|---|
| python-asserts | https://github.com/srittau/python-asserts |
| sqlitedict | https://github.com/piskvorky/sqlitedict |
| ddt | https://github.com/datadriventests/ddt |
| janus | https://github.com/aio-libs/janus |
| ArxivDigest | https://github.com/AutoLLM/ArxivDigest |
| chakin | https://github.com/chakki-works/chakin |
| geotext | https://github.com/elyase/geotext |
| hone | https://github.com/chamkank/hone |
| lice | https://github.com/licenses/lice |
| particle-swarm-optimization | https://github.com/nathanrooy/particle-swarm-optimization |
| readtime | https://github.com/alanhamlett/readtime |
| cachier | https://github.com/python-cachier/cachier |
| munch | https://github.com/Infinidat/munch |
| pandarallel | https://github.com/nalepae/pandarallel |
| sklearn-pandas | https://github.com/scikit-learn-contrib/sklearn-pandas |
| ordered-set | https://github.com/rspeer/ordered-set |

Table 5: URL of Repository.

```
1  {
2  "RepoName": "https://github.com/piskvorky/sqlitedict.git",
3  "FilteredCode": [
4      {
5         "path": "piskvorky_sqlitedict/sqlitedict.py",
6         "content": "1 #!/usr/bin(...truncated)"
7      },
8  "ErrorMessage": "(...truncated) Ran 88 tests in 1.156s
9      FAILED (failures=5, errors=1)",
10 "Issue": {
11 "title": "Autocommit Feature Not Working Correctly and Minor
12     Bugs in Test Cases",
13     (...truncated)
14     },
15 "Patch": "--- a/piskvorky_sqlitedict/sqlitedict.py\n+++
16     b/piskvorky_sqlitedict/sqlitedict.py\n@@ -311,7 +311,8 @@
17     (...truncated)
18 -           self.commit()\n
19 +           if self.autocommit:\n
20 +               self.commit()\n \n
21     (...truncated)",
22  }
```

Listing 1: An example of REPOFIXEVAL