# OpenReview forum: "RepoFixEval: A Repository-Level Program Repair Benchmark From Issue Discovering to Bug Fixing"
_ICLR.cc/2025/Conference — ICLR 2025 Conference Withdrawn Submission_

### Official Review · Reviewer_Vifx · 2024-10-27

**Soundness:** 3
**Presentation:** 3
**Contribution:** 3
**Rating:** 5
**Confidence:** 3

**Summary:**

This paper introduces REPOFIXEVAL, a repository-level Automatic Program Repair (APR) benchmark, which consists of 160 real-world bug fixing suites collected from GitHub projects. The benchmark includes the original buggy programs, associated issue reports, corresponding fixes, and unit tests. Experimental results with 16 LLMs show that the LLMs struggle with generating error messages during the issue discovery phase. Even the best-performing GPT-4o only resolves 12.3% of the bugs. Most LLMs demonstrated performance improvements within a context length of 16K tokens.

**Strengths:**

-	A new benchmark for APR, which contains more types of data (original buggy programs, associated issue reports, corresponding fixes, and unit tests) than exiting benchmarks.
-	Extensive evaluation with 16 LLMs on the benchmark.

**Weaknesses:**

-	The benchmark is not available, which makes evaluation of the work difficult.
-	The Abstract claims that REPOFIXEVAL provides associated issue reports. However, I understand that these issue reports are not reported by users/testers/developers. Instead, they are generated by LLMs.
-	Data quality. It is not clear how good the collected data is. For example, for the data manually identified by five experienced programmers and the issues generated by LLMs, it is not clear the data annotation quality and if all developers agree with the annotation (no Kappa value is reported).
-	The reference issues are used in the inputs to LLMs. What are these reference issues? How are they collected? Why are they needed?
-	Manual data collection effort. The construction of benchmark requires much unnecessary manual effort. For example, five experienced software developers identified fix-related commits from git history. Actually, this can be automated using a mining software repository method such as that described in the following paper: Wu et al., ReLink: recovering links between bugs and changes. In Proc. ESEC/FSE '11, Sep 2011.
-	The benchmark size is small – only 160 bugs are collected from 16 projects. The coverage and representativeness of the bugs remain to verified. Also, only Python program bugs are collected.
-	Presentation issues. There are some presentation issues in the paper. For example:
GPT4-o = > GPT-4o
REPOFIXEVAL utilize => REPOFIXEVAL utilizes

**Questions:**

. Is the benchmark currently available?

. What is the quality, coverage and representativeness of the collected bug data?

. What are the reference issues?

---

### Official Review · Reviewer_Kcb8 · 2024-11-01

**Soundness:** 2
**Presentation:** 3
**Contribution:** 2
**Rating:** 3
**Confidence:** 4

**Summary:**

This work presents RepoFixEval, a benchmark of repository-level real-life software bugs in Python programming language. RepoFixEval, which includes 160 bugs first mines potential bugs from online repositories using commit messages, and human developers extracted actual fixes from commit history. Each instance in the dataset include buggy code, issue reports, fixed code and the corresponding test cases validating the patch. The authors then propose a three-step framework for LLM-based APR, including (1) issue discovery (using LLM as a judge to generate bug issues given repository and failing test results), (2) bug localization (using LLM to pinpoint the location of the bug) and (3) bug fixing (using LLM to consider the generated issue and fix the bug). The study rigorously evaluates 16 LLMs on RepoFixEval, with GPT-4o achieving the best performance by fixing 12.3% of the bugs.

**Strengths:**

- Assessing the performance of LLMs on software engineering tasks in repository-level is valuable. there is not a lot of work addressing repository-level problems
- Large scale evaluation of 16 LLMs
- Evaluation of generated patches in multiple automated steps (segregation and dual testing)
- Ablation study on feeding error message
- Study on context length of models

**Weaknesses:**

- Lack of sufficient background work on APR. The authors did not consider SOTA papers: https://dl.acm.org/doi/abs/10.1145/3650212.3680323, https://dl.acm.org/doi/abs/10.1145/3540250.3549101, https://ieeexplore.ieee.org/abstract/document/10298499
- Relying on LLM for issue discovery
- Relying solely on LLM for bug localization

Some minor issues:
- line 252: amis -> aims

**Questions:**

Q1: The authors have changed the formulation of the Automated Program Repair (APR) problem in Software Engineering. As per (https://ieeexplore.ieee.org/abstract/document/10172803 and https://dl.acm.org/doi/abs/10.1145/3540250.3549101), the APR problem is the second half of debugging (bug localization + bug fix). How do the authors support their claim about this new formulation?

Q2: How the authors support that LLM can be a judge for issue discovery? Did the authors perform a human study on discovered issues to further verify their soundness? Also its unclear to me how "$I_x$ = $M_{eval}(i,P, E)$" is calculated. If the model is prompted to score an issue generated by itself, then there is no reproducibility guarantees. For instance, would the model always give the same score if prompted 100 times?

Q3: Why didn't the authors consider some basic PL/SE techniques when prompting the model to pinpoint the location of the bug? For instance, a subset of the files/code can be eliminated using coverage information, decreasing the search space for the model.

Q4: In APR terminology, patches that pass tests are called plausible patches. A further human check is required to call them "correct" or "successful" patches. Have the authors considered doing this extra step?

---

### Official Review · Reviewer_UuzE · 2024-11-04

**Soundness:** 2
**Presentation:** 2
**Contribution:** 2
**Rating:** 1
**Confidence:** 4

**Summary:**

This paper proposed a program repair benchmark, consisting of 160 python bugs from 16 GitHub repositories and 3 tasks (1) issue generation (2) fault localization, and (3) bug fix. The paper evaluated 16 models and reported that even the best-performing GPT4-o
only resolves 12.3% of bugs. The paper also reports the observations related to context length and task input.

The contributions of the paper cover the following aspects:
- real-world Python project (not novel)
- bugs cross multi-files (not novel)
- 160 bugs: some from git commits and some from injected by developers (the paper is unclear about bug injection amount and process, the dataset of 160 bugs is not large)
- 3 tasks
(1) issue generation from testing results (it can be useful for manual fix, but I don't see why this is a must-step for LLM fix)
(2) fault localization (to fix a bug, any repair benchmark will challenge fault localization of the models)
(3) bug fix
- report the performance and some observations of 16 models (I did not see significant insights reported)

**Strengths:**

+  LLM issue generation from unit testing can be useful for manual/developers' fix.
+  The paper shows that decomposing the APR task into three sub-tasks helps improve performance on bug fix?

**Weaknesses:**

Soundness
- The evaluation for issue generation is not sound:  (1) LLM judge can introduce noise (2)  the metric is proposed by the authors  instead of using standard approaches, e.g., comparing LLM results with existing github issues generated manually.
- It is unclear if 16 models studied are the state-of-the-art. It will be interesting to compare the performance of this benchmark with SWE-BENCH for the models.

Presentation
- Section 3.1 needs concrete details. for example, what types of bugs are injected? how many of 160 bugs are injected?
- Figure 3 is confusing, please improve

Contribution
- The contribution over SWE-Bench is low. The authors claimed that their benchmark has new additions of testing fault localization and LLM's capability of discovering issues itself. However,
(1) if a model can fix a bug, it will need to do fault localization, so all the repair benchmarks implicitly test the fault localization capabilities of a model
(2) When a unit test fails, we can pass it to an LLM automatically via scripts, we don't need LLM for issue generation.

**Questions:**

Does Figure 3 show that decomposing the APR task into three sub-tasks helps improve performance on bug fix?

---

### Official Review · Reviewer_JYht · 2024-11-04

**Soundness:** 2
**Presentation:** 4
**Contribution:** 2
**Rating:** 3
**Confidence:** 4

**Summary:**

This paper proposes a new program repair dataset, RepoFixEval, which targets repository-level bugs. For this dataset, the paper introduces three tasks, i.e., issue discovery, fault localization, and bug fixing, to evaluate the performance of existing LLMs. In addition, the paper conducts solid experiments on popular LLMs to demonstrate their performance on repository-level bug fixing.

**Strengths:**

+ The writing in this paper is clear and easy to understand.

+ The experiments are solid. This paper evaluates several popular LLMs and includes a detailed analysis of experimental results.

**Weaknesses:**

1. Why not Defects4j?

Defects4j is a long-standing repo-level bugs, which has been well used in the AI/SE community. Given the justification, I do not see why they cannot be used? It seems that the authors are re-inventing the wheel.

The reviewers also encourage the authors to check Defects4j, and illustrate how your dataset is different, except for programming language difference


2. A large number of single-file patches in the dataset, are they file-level or repo-level?

The paper does not provide sufficient evidence to show that the proposed dataset is genuinely repository-level. If I understand correctly, the “Patch” field in each “file_xxx.json” in the supplementary material is the ground patch. However, in the supplementary material, out of all 140 bugs, there are 109 where the ground truth patches contain only one file modification, which may not be considered repository-level bugs, as indicated in Appendix A.2. Also, in Table 2, the dataset statistics of “# File” is only 1.91, which is less than two, meaning that for more than a half bug cases, the “# File” is only 1.

3. The quality of the dataset.

The paper states that RepoFixEval consists of 160 **real-world** bug-fixing instances. However, in Line 171, it mentions that some bugs are manually injected. The proportion of injected bugs is not reported. This raises concerns about dataset quality control. The paper should explain how it ensures that manually injected bugs behave like real-world ones.

**Questions:**

1. Why not Defects4j?
2. Is there any clarification available regarding the inclusion of bugs that involve only a single file modification?
3. Could the number of manually injected bugs in the proposed dataset be specified, and could further details on how dataset quality is controlled be provided?

---

### Note · Authors · 2024-11-15

**Comment:**

Thanks for the efforts of ACs. Thanks for the valuable reviews of all reviewers.

**Withdrawal Confirmation:**

I have read and agree with the venue's withdrawal policy on behalf of myself and my co-authors.